# "If I get sick here, I will never see my children again": The mental health of international migrants during the COVID-19 pandemic in Chile

**Alice Blukacz**[1], **Báltica Cabieses**[1]*, **Alexandra Obach**[1], **Paula Madrid**[1], **Alejandra Carreño**[1], **Kate E. Pickett**[2], **Niina Markkula**[3]

**1** Facultad de Medicina Clínica Alemana, Instituto de Ciencias e Innovación en Medicina, Universidad del Desarrollo, Santiago, Chile, **2** Department of Health Sciences, University of York, York, United Kingdom, **3** Faculty of Medicine, Department of Psychiatry, University of Helsinki and Helsinki University Hospital, Helsinki, Finland

\* bcabieses@udd.cl

## Abstract

### Background

The COVID-19 pandemic has had an impact on the mental health of international migrants globally. Chile has managed its response to the pandemic in an ongoing context of social unrest and combined regional migratory and humanitarian crisis. The country's population presents a high prevalence of common mental disorders and a high suicide rate, with limited access to mental healthcare. International migrants in Chile represent 8% of the total population, and although a socioeconomically heterogenous group, they face social vulnerability, a range of mental health stressors and additional barriers to access mental healthcare. This study describes the mental health outcomes, stressors, response, and coping strategies perceived by international migrants during the COVID-19 pandemic in Chile.

### Methods and findings

A qualitative case study was carried out through individual online interviews to 30 international migrants living in Chile during the pandemic and 10 experts of the social and health care sectors. An inductive content analysis was carried out, a process during which the researchers sought to identify patterns and themes derived from the data. Participants experienced mainly negative mental health outcomes, including anxiety and depression symptomatology. Stressors included the virus itself, work, living and socioeconomic conditions, discrimination, fear for their family and distance caring. Institutional responses to address the mental health of international migrants during the pandemic in Chile were limited and participants relied mainly on individual coping strategies.

### Conclusions

The pandemic can represent an important opportunity to strengthen mental health systems for the general population as well as for population groups experiencing social vulnerability,

**Data Availability Statement:** The data set is available through the following doi: 10.6084/m9.figshare.21493548.

**Funding:** BC received funding from Dirección de Investigación y Doctorados (DID), Universidad del Desarrollo Chile, project supported by ANID COVID fund, COVID0873, Chilean Government. https://www.udd.cl/investigacion/fomento-a-la-investigacion/ https://www.anid.cl The funders had no role in study design, data collection and analysis, decision to publish, or preparation of the manuscript.

**Competing interests:** The authors have declared that no competing interests exist.

if the issues identified and the lessons learned are translated into action at national, regional, and international level. Promoting the mental health of international migrants means recognising migration as a social determinant of mental health and adopting a cross-cultural as well as a Human Rights approach.

## Introduction

The COVID-19 pandemic has caused a threefold health crisis, the direct threat brought by infection and the overburdening of health services, the indirect, long-term threat caused by delayed care for chronic and non-communicable diseases [1,2], and the impact on the social determinants of health related to the disruptions of daily activities caused by nonpharmaceutical prevention, revealing and exacerbating socioeconomic inequalities around the world and especially in countries with insecure job markets and fragile or insufficient social protection systems in the Americas [3,4].

More specifically, the consequences of the COVID-19 pandemic on the mental health of the general population, as well as of population groups who may experience specific instances of social vulnerability, such as international migrants, have been highlighted in the existing literature globally [5,6]. Although there is no systematic evidence regarding the higher prevalence of mental health issues among international migrants [7,8], migration is defined as a social determinant of mental health, as international migrants, including the forcibly displaced, asylum seekers, refugees and labour migrants, face stressors during all the phases of the migration cycle, potentially altering mental health outcomes [7,9]. Not all international migrants experience social and health vulnerability, however, an important proportion may be at higher risk than the general population, as they face difficulties to regularize their migratory status, precarious employment, limited access to health services, insufficient social protection, as well as exclusion, discrimination, xenophobia, violence and abuse in countries of transit or settlement [10]. Additionally, acculturation processes can have an impact on mental health outcomes [11,12].

In the context of the pandemic, in terms of symptomatology of mental health issues in international migrants, the World Health Organization (WHO) ApartTogether Survey found that most of the participants reported feeling depressed, worried, anxious, lonely, angry, stressed, irritated, hopeless, having more sleep related problems and using more drugs and alcohol than before the pandemic [13]. Qualitative and quantitative studies from several countries show similar results. In the US, different studies focusing on the mental health of international migrants have found increased substance abuse and suicide ideation [14,15], worsened anxiety and depression levels as well as high levels of psychological distress [16] as a result of the pandemic. In Korea, a high prevalence of severe anxiety disorder was reported during the pandemic among international migrants [17]. In Switzerland, the COVID-19 pandemic brought increased sources of preoccupations to undocumented migrants and migrants undergoing status regularization, resulting in poor psychological health [18] and in Spain international migrants presented worse mental health outcomes than the Spanish-born, with refugees presenting the worst scores [19]. Finally, In Chile, in the first month of the pandemic, according to an opinion poll conducted with 1650 international migrants, 90% reported feeling anxious and 73% sad or depressed because of the pandemic [20]. Another, qualitative, study highlighted that international migrants from Venezuela entering Chile via unauthorized crossing-points after travelling by foot for several weeks presented mental health issues [21]. The factors identified for increased mental health issues during the pandemic are the following: precarious or worsening socioeconomic conditions, pre-existing mental health issues,

migratory status, unemployment, loss or decrease of income, food and housing insecurity, uncertainties around access to healthcare, fear of infection, knowing someone infected, misinformation and social isolation [16–18,20,22–24].

Chile received the COVID-19 pandemic in an ongoing context of social and political turmoil as the riots that started in October 2019 represent the biggest political crisis since the return to democracy in 1990 [25]. Grievances were focused on socioeconomic inequalities, unequal access to healthcare, retirement systems, indigenous peoples' rights and reparation, and human rights [26–28]. After months of riots violently repressed by the police and military forces [29], the pandemic put a halt to demonstrations of discontent. Although important reforms are underway, mainly through the rewriting of the Constitution, structural and pervasive socioeconomic grievances have only partially been addressed by the government of President Sebastian Piñera, and living conditions worsened for the most vulnerable as the pandemic progressed [30].

With regards to mental health, Chile has a high prevalence of common mental health disorders at national level, as 23.2% of the years of life lost because of disability or death are due to neuro-psychiatric conditions [31] and the suicide rate is 10.7 per 100,000 inhabitants [32]. Affordability, availability, accessibility, and acceptability of mental healthcare services are hampered by the segmented nature of the Chilean healthcare system, with an underfunded and overcrowded public sector unable to respond to the needs of its beneficiaries and a private sector reluctant to include mental healthcare in its coverage plans [33,34]. Notwithstanding this unfavourable context, mental health has been increasingly put at the top of the public health policy agenda and the first Mental Health Law was passed in May 2021.

In the last decade, Chile has been receiving an increasing number of international migrants, who now represent about 8% of the total population or nearly 1,500,000 people [35]. Migrants from Venezuela represent a third of the total foreign population, followed by migrants from Peru, Colombia, Haiti and Bolivia [35]. International migrants in Chile, on average, have higher educational levels and employment rates than nationals, however, they face higher rates of multidimensional poverty [36]. With respects to health insurance coverage, international migrants with resident status are eligible for both public and private coverage and undocumented migrants are guaranteed basic public health insurance [37]. According to data from the brief 2020 National Socioeconomic Characterization Survey (*Caracterización Socioeconómica Nacional–CASEN*), 12% of international migrants reported not being affiliated to any health insurance scheme, a rate more than 3 times higher than nationals and 74% were beneficiaries of the public health system, 12% of the private system and 2% to the military health insurance system [38].

The mental health of international migrants, refugees and asylum seekers in Chile is a topic of growing interest and they are included in the National Mental Health Plan 2017–2025 as a population group of interest considering migration as a social determinant of mental health [31]. The International Migrant Health Policy launched in 2018 calls for the inclusion of international migrants into strategies, programmed and interventions aimed at promoting mental health [39]. However, the social and political context has grown increasingly hostile to migrants in recent months, adding to the already adverse conditions brought by 2019 social unrest and COVID-19 pandemic. The exodus from Venezuela coupled with the visa requirement for Venezuelan nationals to enter Chile and the closing of the border in the context of the pandemic, has led to an increasing number of Venezuelans entering the country through unauthorised crossing points in Northern Chile, with little option to obtain a residency permit and work in the formal labour market [21,40]. This migratory and subsequent humanitarian crisis has been instrumentalised in the context of the 2021 presidential elections and violent demonstrations of xenophobia have taken place [41].

Chile's recently arrived and largely discriminated migrant population has faced the pandemic in a context of poor access to health services and social turmoil in the country. In other countries, international migrants have faced negative mental health outcomes and a range of stressors. In this context, the objective of this study is to describe the mental health outcomes, stressors, institutional response, and coping strategies perceived by international migrants during the COVID-19 pandemic in Chile, based on qualitative interviews conducted in November and December 2020.

## Methods

This methods section is structured following the consolidated criteria for reporting qualitative studies (COREQ) 32-item checklist [42].

### Theoretical framework

A collective case study was carried out under a qualitative paradigm, where the researchers sought to analyse the experiences of international migrants and health and social experts with regards to mental health during the COVID-19 pandemic in Chile. This study design was selected as it allows for an in-depth understanding of the participants' experiences, their context, and processes [43]. This analysis is a secondary analysis with a focus on mental health, as the primary study focused on the social vulnerability and resources of international migrants during the COVID-19 pandemic [44]. Although the initial study did not specifically seek to explore the mental health of international migrants during the pandemic, mental health emerged as an important aspect of the experience of the participants interviewed with regards to COVID-19. In that sense, the authors made the decision to carry out a separate, in-depth analysis specifically focused on mental health, considering the relevance of the topic as highlighted in the literature reviewed in the introduction. According to Heaton, carrying out an additional in-depth analysis, or supplementary analysis, allows for a "more in-depth analysis of an emergent issue or aspect of the data, that was not addressed or was only partially addressed in the primary study" [45]. The question guiding this secondary analysis is the following: what are the mental health outcomes, stressors, institutional response, and coping strategies perceived by international migrants during the COVID-19 pandemic in Chile?

### Participant selection and recruitment

A total of 40 people participated: 30 international migrants living in Chile during the COVID-19, and 10 health and social experts who worked with international migrants before and during the pandemic. Participants were recruited and data was collected during November and December 2020, when several COVID-19 restrictions were still in place (localised lockdowns, curfew, among others) and vaccination was not yet available to the general public. The sampling was purposive, as researchers strove to achieve diversity and representativeness of experiences and discourses by including participants with different migratory trajectories, migratory status, employment status and occupations, educational levels, health insurance schemes and countries of origin. Gender representation was also taken into account. Specifically, we sought to have representation from the five main countries of origin of international migrants in Chile, which are Venezuela, Peru, Haiti, Colombia, and Bolivia, as well as other countries such as Argentina and Brazil. In terms of representation of different age groups, we focused on international migrants under 45 years old, who represent almost 80% of migrants in the country, but also interviewed participants over that age, in order to include the experience of international migrants in a different stage of their life cycle. With regards to migratory status, we sought to represent migrants in different situations based on feasibility, taking into account

that people in a precarious or irregular migratory situation may be harder to reach. Healthcare coverage was also considered, based on the results of the CASEN Survey 2017, where 16% of all international migrants reported not being covered and among those covered, 80% reported affiliation to the public system and 20% to the private system [36]. Finally, the study focused on the Metropolitan Region of Santiago and the Northern regions of Arica y Parinacota and Antofagasta, as these two regions concentrate over 75% of the total migrant population in the country [36].

Inclusion criteria for international migrants were the following: being foreign-born, over 18 years-old, Spanish-speaking, living in Chile during the COVID-19 pandemic, living either in Santiago or any of the Northern regions. Inclusion criteria for experts were as follows: being over 18 years-old, Spanish-speaking, having worked with international migrants or with a specific focus on international migration either as a healthcare professional, social worker, non-governmental organisation worker or government worker, before and during the COVID-19 pandemic. No specific additional exclusion criteria were applied for either group.

Recruitment of participants took place entirely remotely, considering the social distancing and travel restrictions in place in Chile at the time due to the pandemic. International migrants were primarily recruited among the researchers' extended networks and with the support of one of the largest non-governmental organisations working with migrants in Chile, *Servicio Jesuita a Migrantes*. Snowball recruitment was carried out to reach the target sample. Potential participants were first contacted by phone, and none refused to participate. Three people failed to answer the call set up for the interview and did not reschedule, mentioning lack of time as a reason. Experts were recruited via email among professional networks of the researchers.

## Setting

Individual semi-structured interviews were conducted, each of approximately 45 minutes, through Zoom or WhatsApp videocall by an experienced researcher trained in the technical and ethical aspects of conducting qualitative research. Participants usually took part in the interview from their home, alone or with family members in proximity. Experts sometimes participated from their workplace. Each participant only participated in one interview.

## Data collection

The interviewer used the interview guide in S1 File. The guide was prepared by BC and AO. Interviews were recorded for transcription. The interviewer did not take field notes and transcripts were not shared with the participants for correction due to the short timeframe for data collection. Data saturation was discussed among the research team, and it was agreed to have been reached after 40 interviews with regards to the main themes of the primary study: experiences of social vulnerability during the pandemic and strategies and resources to cope.

## Data analysis

All interviews were fully transcribed by the interviewer and the data was analysed by AB and PM. An inductive content analysis was carried out, a process during which the researchers sought to identify patterns and themes derived from the data. Considering that this is a secondary analysis, a new codification process was carried out with the data identified as relevant for mental health in the initial analysis. As the interviews carried out were semi-structured, the dataset obtained was comprehensive enough, thus avoiding the "problem of data fit" [45]. With a specific focus on mental health, an open coding process was undertaken and main categories and respective subcategories were generated [46].

As the researchers carried out the process separately, a process of consensus-building among both researchers was carried out, where they discussed any discrepancy in the generation of categories and agreed on a final version.

With regards to reporting the results, we first present the coding tree (Fig 1 in results section) in order to provide an overview and then present a detailed analysis of each emerging subcategory, together with at least a quote for each. Quotes were selected according to the following criteria: illustrative (provides an explicit example), succinct, and representative (expresses efficiently and faithfully the pattern described) [47]. Whenever quotes were shortened by removing irrelevant material to ensure succinctness, an ellipsis "(. . .)" was used, and authenticity was always prioritised in doing so.

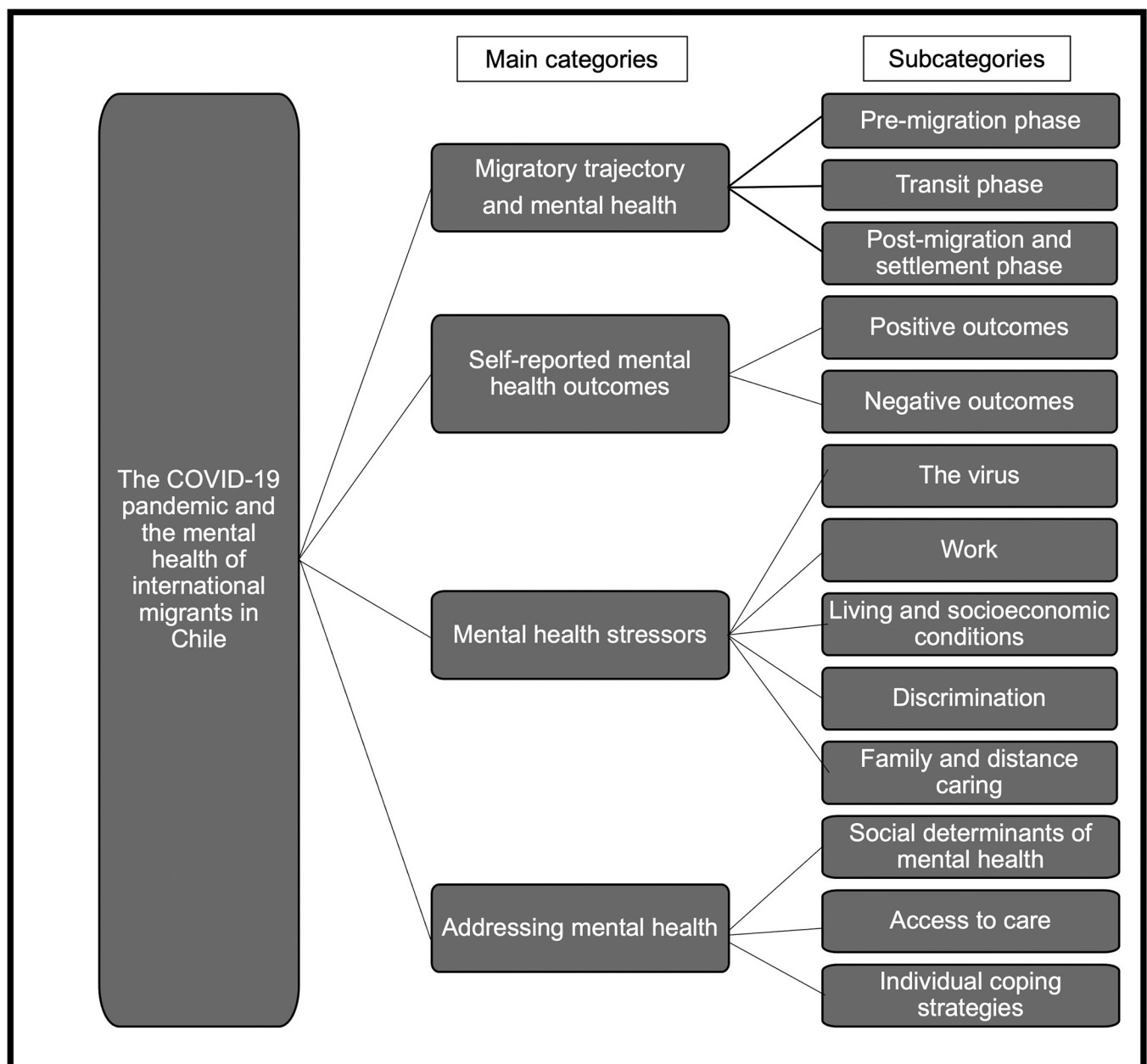

**Fig 1. Coding tree.**

## Research team and reflexivity

The research team is multi-disciplinary, including health professionals and social epidemiologists, a social scientist and two anthropologists specialised in health studies. All had previous experience carrying out research with international migrants and one of them is herself an international migrant in Chile. Although recruitment of participants was carried out among the researchers' extended networks, none of the interviewees had a previous tie or relationship with the interviewer.

## Ethics

The study was carried out in accordance with the relevant guidelines and regulation for research involving human beings, including the Declaration of Helsinki and was approved by the Ethics Committee of the Universidad del Desarrollo before the start of fieldwork. Participation was voluntary and participants filled an informed consent form available online through Google Forms before taking part in the interview, thus written informed consent was secured. Additionally, they could withdraw from the study at any point and refuse to respond any of the questions. The consent form included key information on the project and guaranteed that participating would not have an impact on the participant's relationship with any of the parties involved. Additionally, it informed participants of their right to withdraw from the project at any stage. Finally, it informed participants that although they would not directly benefit from participating, their participation would be of indirect benefit to the migrant community in Chile. All data were recorded anonymously and no information allowing to identify the participants was kept except for the consent forms which are held in the PI's computer in a locked file. With regards to the ethical aspects of carrying out a secondary analysis, the research team that carried out the initial study is the same as the one who carried out this supplementary analysis and no confidential data was shared with external researchers. Additionally, second analyses are within the scope of the outputs stated in the informed consent form.

## Results

We interviewed 30 international migrants from Latin America and the Caribbean: 23% (7) were from Venezuela, 17% (5) from Peru, 17% (5) from Colombia, 10% (3) from Haiti, 7% (2) from Bolivia, 7% (2) from Ecuador, 7% (2) from Argentina, 7% (2) from Brazil, 3% (1) from Cuba and 3% (1) from Uruguay. The majority, or 73% (22), lived in the Metropolitan Region of Santiago, 20% (6) in the Arica y Parinacota region and 7% (2) in the Antofagasta region. With regards to gender, 57% (17) identified as female and 43% (13) as male. 40% (12) of the participants were between 25 and 29 years old, 30% (9) were between 30 and 35 years old, 13% (4) reported being between 36 and 40 years old, 10% (3) between 41 and 45 and finally 7% (2) was over 45 years old. Migratory status is another key characteristic and 13% (4) reported an irregular migratory status and 37% (11) reported being in the process of obtaining either a temporary or permanent residence permit, meaning that half of the participants presented some degree of migratory instability. Furthermore, 23% (4) of participants reported arriving in Chile between 6 months and 1 year prior to the interview, 60% (18) had been in the country for 1 to 5 years and 7% (2) between 6 and 10 years. 20% (6) reported living in Chile for over 10 years. The majority of participants, or 77% (23) had worked either formally or informally during the week in which they were interviewed. Finally, 63% (19) of the participants reported public health coverage (FONASA), 17% (5) reported private coverage (ISAPRE), 17% (5) reported not being covered or not knowing and only 3% (1) reported having an international healthcare insurance.

Among the 10 experts, 60% (6) identified as male and 40% (4) as female; and 50% (5) worked in the social sector and 50% (5) in the health sector. It is important to highlight that two of them were themselves international migrants from Venezuela and Haiti.

Fig 1 describes the main categories and subcategories from the analysis process as described in the methods section.

## Background: Migratory trajectory and mental health

Considering that migration is a social determinant of mental health, it is important to first present and describe the migratory trajectories of participants and their relation to mental health. The purpose of this background section is to contextualise the specific mental health outcomes and stressors described during the pandemic.

## Pre-migration phase

Looking at the reported mental health outcomes of participants during the pre-migration phase is important to address mental health from a lifecycle perspective. When describing their migratory trajectory, some participants mentioned life-changing events and mental health issues as connected to the making the decision to migrate:

> "I came here to Chile more than anything because of that depression I had, I had another daughter who died in Peru, so I was very depressed, and I wanted to clear my mind because I felt guilty, so I came here"

> Peruvian woman, Arica (1)

Other participants described situations of violence and persecution in their pre-migration phase, which may be factors of long-term stress and trauma. One of the participants described having suffered persecution and extorsion from armed groups in Colombia, leading to the decision to seek asylum in Chile. Another recalled having suffered gender based and intrafamilial violence:

> "I came here on my own, I did not know Chile, I came in 2010, because I had my son, he is 13, I had him when I was 14, because I did not know anything, I mean, his dad used to beat me up, to abuse me, he did not let me leave the house and I went through so many things, and at that point no one helped me, not even my siblings, they turned their backs on me and everyone helped him out. This is why I decided to live far from them, that is why I came here."

> Bolivian woman, Arica (1)

Here, mental health can be explicitly identified as a reason for migrating and closely linked to the process or reasons for migrating can be linked to events that can have a mental health impact throughout the migration cycle. These elements were reported especially by women facing socioeconomic vulnerability both in their country of origin and in Chile.

**Transit phase.** The transit phase of the migratory cycle may also have on impact on the mental health outcomes of migrants. Some participants travelled to Chile by plane and did not highlight it as a particular element worthy of attention in their migratory trajectory. Difficult transit conditions can be stressors and alter mental health, and other participants reported having travelled by land for many days or months, at times by foot, with young children and

facing misinformation and fearmongering with regards to what to expect upon arrival to Chile. Several entered through non-authorised crossing points:

"And so, we spent 12 days altogether, 7 days to get to Chile, that were very hard, very hard, because we travelled with four children, three of them very young and a teenager, it was very difficult."

Cuban woman, Antofagasta (1)

"Yes, we walked all the way here. We came with our children so we had to stop several times and it took us a month (. . .) they told me, if you go to Chile, if you cross the Chilean border, they will take away your children they will kill you and your wife, and they will enslave your children (. . .) it is very dramatic and scary but then with time, you wish to have your family here, you wish things were different but it all depends on having your (visa)"

Venezuelan man, Arica (1)

Although the participants did not necessarily explicitly link their transit conditions and with their mental health, some of the experiences reported can reasonably be expected to be stressful, and the way in which participants recount them ("very hard", "very difficult", "scary") gives us some clues as to the potential impact this may have on their mental health.

**Post-migration and settlement phases.** The post-migration and settlement phases of the migratory cycle are particularly relevant to mental health outcomes as dynamics linked to existing patterns of class-based discrimination and racism are reproduced in the lived experiences of international migrants in Chile. Additionally, and more specific to the context of migration and cultural diversity, although tightly related to such existing patterns of discrimination, processes of acculturation have an impact on mental health.

First, the (mis)management of migration, especially the ordeal of getting a visa application approved–due to lack of reliable information and long waiting times, creates uncertainty and spaces of social vulnerability with consequences in the short- and long-run on general wellbeing, living conditions and employment. Many participants encountered issues in trying to regularise their migratory status regardless of their background and reason for migrating and consequently expressed some degree of powerlessness as a result:

"Can you imagine, there was a time, a time where I was in limbo, unable to do anything"

Colombian woman, Santiago (2)

Second, migratory regularization raised issues to most participants, and even more so to those who entered Chile via non-authorised crossing points, leading to consequences on their possibilities to obtain a residence permit and to work in the formal sector. This, in turn, led to employment being a key topic among all participants in their migratory trajectory, as they described difficulties to find work as a foreigner, having to work in unqualified jobs despite being highly educated and suffering discrimination and abuse, with an impact on wellbeing and self-esteem:

"Yes, I used to get in at 8am sharp but I got out at 10pm instead of 6pm, I always stayed, they always made me do laundry until late, every day I went to work and once they did not pay me, they refused, and (my boss) was mean, she used to make me cry a lot"

Bolivian woman, Arica (1)

Third, participants also suffered discrimination in other contexts, and for instance one participant describes moving to Chile from Argentina as a child and facing bullying at school as a foreigner:

"of course, the first years were very hard, I remember not wanting to go to school, crying at the door because the other children bullied us because of how we spoke, our accent, things like that, but then we started to get used to it"

Argentinian woman, Santiago (1)

Discrimination, however, was not reported by all the participants, and was usually connected to heightened perceptions of difference, such as skin color or a strongly different accent, and other participants identified discrimination as something experienced by "others", more "different" than themselves:

"Of course, I am a white girl with an undefined accent, so, you see, I mean in my case I cannot complain, I have never experienced discrimination, they treated mi very well but I think this has to do with my condition, the way I am (. . .) maybe if I spoke like a Colombian, for instance, if my accent were typically Colombian, I think maybe that could be an issue, if my skin colour were different, I think that could be another issue.

Colombian woman, Santiago (2)

Finally, some participants reported experiencing feelings and symptoms akin to migratory grief, or a feeling of loss associated with the migratory process, as well as, in some cases, difficulties linked to cultural differences:

"My first months here were extremely hard, because I got kind of depressed, I did not want to leave the house, I missed my children a lot, my mother, my grand-mother, so I just spent my time in bed crying, but then after about four months I had to make a decision, look for a job and get ahead, or leave, but I could not afford to leave at that point"

Colombian woman, Santiago (3)

"I do not mean to offend you, but Santiago was very difficult because people are more distant (. . .) in my country, being pretty blunt is a cultural characteristic, it has good and bad sides, but this is how I prefer it"

Uruguayan man, Santiago (1)

International migrants in post-migration and settlement phases can face a range of stressors linked to legal processes for migratory regularization, precarious employment usually because of irregular migratory status, discrimination in different settings, as well as migratory grief and acculturation difficulties. Although experiences can be very different depending on personal history in the country of origin, during transit and once in Chile, the stressors are connected to wider dynamics of racism, classism, gender violence, armed conflict affecting civilians, all of which sometimes intertwine, translating into migratory precariousness created by legal and bureaucratic barriers to timely migratory regularization, informal employment and abuse from employers towards international migrants, as well as general patterns of discrimination towards foreigner perceived to be visibly different. Additionally, migratory grief and acculturation processes as stressors can occur as part of the migration process, with different consequences on mental health at least in the short run.

Presenting results identifying the different stressors emerging from the participants' migration "story" contributes to establishing migration as a social determinant of mental health and sets the background before analyzing the mental health of international migrants during the pandemic.

## Self-reported mental health outcomes during the pandemic

**Positive outcomes.**   Throughout the interviews, participants reported a range of mental health outcomes, either positive or negative, connected to the pandemic. Self-reported positive mental health outcomes included feelings of happiness and overall satisfaction and wellbeing, mostly among the more highly educated, financially stable participants who entered Chile via regular routes, and who, in some case, had access to positive coping mechanisms:

"I love life, the time I have at home, I like to enjoy it and I always move things around in the house, because apart from being a fashion designer, I am an art professor and many other things, so I painted, I did about 6 new courses and other things I wanted to do, I saw every show on Netflix"

Argentinian man, Santiago (1)

"They should put more emphasis on mental health above anything else. Because it is very important, I mean, during the pandemic, I can assure you I would have had a thousand crises. I suffer from anxiety and depression, but I had it all under control."

Peruvian woman, Santiago 1

In that sense, only participants experiencing a lower degree of social vulnerability and access to mental healthcare according to their needs reported positive mental health outcomes during the pandemic, although, importantly, not all of them did, suggesting limitations in these protective factors.

**Negative outcomes.**   The majority of participants reported symptoms potentially associated with negative mental health outcomes and mood disorders such as depressive and anxiety disorders. The reported symptoms included feeling sad, scared, overwhelmed, irritable, anxious, and having trouble sleeping, even using vocabulary pertaining to formal mental health diagnosis to describe how they felt:

"I have been feeling downhearted and I used to get anxious, and not being able to sleep, but now things have been a bit calmer. I feared getting infected and also I did not know when all of this would pass, because after a while money gets tight and if you do not have a job. . . what else can I do here alone with my family. . ."

Ecuadorian woman, Santiago (1)

As pointed out in the previous section, negative outcomes were reported by participants of all background and socioeconomic position, and it is thus necessary to examine the different health stressors identified in the interviews.

## Mental health stressors during the pandemic

Several different, albeit usually interrelated, elements were identified by the participants as stressors during the pandemic, some of them related to migratory trajectory and status.

**The virus.** The first main stressor was the virus itself or the fear of being infected, when the risk perceived was high:

"We never went out, we did not even lean out of the window because as it happens, we live close to a nursing home and you can imagine how traumatizing it was for us when the residents started getting infected, and if the cat came in, we got scared the cat would bring in the infection, the virus, that thing, it was crazy"

Venezuelan woman, Santiago (3)

The fear of infection sometimes triggered other fears specific to dimensions of being a migrant, such as limited support networks, being far away from family and not being able to see them again, or not knowing if medical attention would be provided as a foreigner or because of their migratory status, which had repercussions on participants' behaviour:

"I have been feeling very panicky, I felt fear, because I thought, if I get sick here, I will never see my children again, so I did not go out at all unless it was to go to work, not even across the street"

Colombian woman, Santiago (3)

Another aspect of the virus being a stressor, was knowing someone infected or who died because of the virus. For instance, a participant described having trouble dealing with the death of his young cousin and another one with having her father ill in Venezuela:

"My dad got COVID in Venezuela, and that was. . . I mean I do not wish it upon anyone. It was despairing, I mean being away from my family and not having the money either (to help them)"

Venezuelan woman, Santiago (2)

Stress due to the virus was thus two-fold: getting infected or family member being ill or having died because of the virus. Although this is expected to be a very common stressor among the general, non-migrant population, in some cases, it was also related to aspects of being a migrant, usually with regards to being away from family members, or the precariousness of their status in the country and fear of not being able to access medical care should they need it.

**Work.** Work and employment were also identified as stressors by participants, although for a range of different reasons. The first one is loss of employment and the subsequent loss of income:

"I have been very anxious, more than anything because I did not know what was going to happen, or what was coming next or what we were going to do, not being able to plan anything into the future, this really affects me and also being unemployed is horrible, it is very overwhelming, what can I say, so many different factors, not only work but also my family, economically. . ."

Venezuelan woman, Santiago (2)

Conversely, other participants experienced stress related to not being able to stay home from work or work from home during lockdowns, either as informal workers due to obstacles

to regularise their migratory status, implying having to break lockdown rules and risk a fine to earn their income or as formal workers in essential sectors:

"(the police) stopped me and I told them, you will have to forgive me (. . .) I am not in my own country, but if you had three kids at home you would also have to come out to work because if your child tells you, dad, I am hungry or dad I want to eat this or that, what should I tell them? Wait until the lockdown is lifted?. . . They understood and let me go."

Venezuelan man, Arica (1)

More specifically, worsened working conditions were also reported with mental health implications, as mentioned for instance by a live-in domestic worker who was no longer allowed to leave her workplace when the pandemic began and saw her workload increase:

"When COVID started, I was at work and I was working as a live-in nanny and they did not allow me to go out anymore, they said, you have to just stay in. And that was really hard for me, because I was starting to get more comfortable, to do things outside of work, and not allowing me to go out of the house anymore was very hard (. . .) I had to sanitize the food, to cook all day (. . .) that really threw me off and I had to quit, it was all wrong"

Peruvian woman, Santiago (2)

It is likely that work and employment were also stressors for the local population, considering the important disruptions to daily activities raised by lockdowns and other mobility disruptions put in place to slow contagion, however, some international migrants face additional stress linked to working in the informal sector with daily wages and no social security, due to their irregular migratory status. Furthermore, although employment conditions may have worsened for the local population experiencing higher degrees of social vulnerability, migrant women are more likely to be working in worse conditions in the first place, and face even worsened conditions as a result of the pandemic, both in terms of the work itself and poor job security. In that sense, work and employment were important stressors for international migrants facing migratory precariousness or inserted in precarious and/or informal sectors, with a significant risk of sudden loss of income or exposure to the virus as well as fines for breaking lockdowns.

**Living and socioeconomic conditions.** Some participants saw their living and socioeconomic conditions worsen as a result of the pandemic, with increased anxiety around loss of income or inadequate housing during lockdowns:

"we moved to another room, I mean a shared flat, and I am not going to lie, it is horrible, it is horrible because there is no privacy (. . .) and also being inside all day in such a small space, because I know I suffer from asthma so I am very careful, I almost never went out during the pandemic, I mean never ever, but that also takes a toll, psychologically, being in all day, I am sure it has collateral effects."

Venezuelan woman, Santiago (2)

Also linked with employment, suffering a change in living conditions can be a stressor, and although it is not exclusive to the migrant populations, as described earlier, international migrants face increased work precariousness and thus may be at higher risk of negative changes in their living conditions during a crisis such as the pandemic, which may be exacerbated as well by limited support networks.

**Discrimination.** Although patterns and dynamics of exclusion and discrimination underlie other stressors, direct discrimination was also brought up by a participant with relation to the Haitian community in Chile:

> "Right at that time the news came out I think in Valparaíso, I do not remember where, that a Haitian had COVID, they accused him of having COVID and then they started shoving him, I saw the video, and after accusing him, they had him take a test, he did not have COVID so now all the Haitians who saw the video are saying that they are accusing us because we are Black, because we are Haitians"

> Haitian man, Santiago (1)

It is important to note that discrimination against international migrants and the Haitian population in particular did not arise as a result of the pandemic, and participants indeed reported having suffered it before the pandemic as migrants in Chile. However, in some specific instances, tensions arose with regards to the alleged responsibility of members of the Haitian community for the spread of the virus, something that is rooted in existing racist dynamics in the country. Although the participant reporting this instance did not specifically mention it with regards to his mental health, these demonstrations of racism and physical violence may have acted as stressors for members of the Haitian community in the country.

**Family and distance caring.** Finally, a key aspect of migration is being separated from family and loved ones, an aspect exacerbated by border closures and mobility restriction during the COVID-19 pandemic, leading to feelings of isolation and loss of key life events, as told by some participants:

> "We were very used to travelling back very often and now we cannot, for instance my parents have not been since Christmas, almost a year now. I mean, videocalling has saved our lives, but for instance, during the pandemic two of my nieces were born, and we are very close to our family, so we have been struggling with this situation"

> Argentinian woman, Santiago (1)

> "One of the things we have observed has to do with the mental health of migrants, with the fact that they are actually experiencing a pandemic, away from their family and away from their support networks"

> Expert 3

Additionally, leaving children behind may mean experiencing "distance worrying" and performing "distance caring", leading to feeling of sadness and powerlessness, as expressed by a participant with teenage children in Colombia:

> "It has been hard, also, I used to call my children and tell them, please do not go out, take care of yourselves, and then I would talk to my eldest and he would say that his brother was out playing football, and I would say, please be careful, you should not be going out. (. . .) That is why I have been scared, not so much for me but for my children, because I am not there (. . .) I have not feared so much for myself, I am careful, but them. . . so it has been a bit sad not being there."

> Colombian woman, Santiago (3)

In that sense, being a migrant during a global pandemic implies an additional distancing brought about by the impossibility to travel abroad and although these restrictions have since been lifted, at the time the interviews were conducted, they were very much still in place in Chile with no clue as to when they would end. Participants expressed distress when mentioning that topic, adding to all the other stressors identified.

## Addressing the mental health of international migrants during the pandemic

Considering the symptomatology and the different stressors identified throughout the interviews with international migrants, it is important to describe results related to, on the one hand, the institutional approach to social determinants of mental health and access to mental healthcare, and on the other hand, individual coping strategies.

**Institutional gaps: Social determinants of mental health and access to mental healthcare.** The experts interviewed pointed out to the lack of an adequate response to the mental healthcare needs of international migrants, in terms of cross-cultural relevance and addressing the social determinants of mental health from the healthcare sector:

"Depending on the different cultures, there are a lot of issues associated with mental health that cannot be resolved in a healthcare centre, nor with a drug, seeing a psychologist or a psychiatric hospitalisation. There are other means, at family, social or community level, that can be used to buffer, to protect, like a mattress that can be used to address mental health issues"

Expert 4

The international migrants interviewed mostly focused on barriers to access, mainly cost and lack of availability:

"It is almost a privilege to access mental healthcare"

Peruvian woman, Santiago (1)

"Yes, I have had healthcare needs, especially mental health. No, I have not been to the psychologist because I am always busy, the only free time I have is at night, from 11pm or sometimes during the weekend, but there is no psychologist available then."

Peruvian man, Santiago (1)

Among all participants, only three used mental healthcare according to their needs during the pandemic, two of them continuing an existing therapy and another one relying on online therapy with psychotherapists in her home country:

"I was in therapy all year, because otherwise I would have collapsed and also because really, I have colleagues in Venezuela who can treat me online for free"

Venezuelan woman, Santiago (2)

Considering the needs for mental healthcare and the institutional gaps identified, civil society organisations have developed temporary solutions to link international migrants to mental healthcare professionals during the pandemic:

"we developed partnerships with networks of volunteer psychologists and psychiatrists to refer many of our beneficiaries, for them to receive support during the crisis (. . .) they arrive scared and overwhelmed, so we had to find ways to address their mental health"

Expert 3

These results point towards important structural gaps within the healthcare system to address the mental health of international migrants, some of which, such as cost and hour limitations may be shared with the local population. Although these gaps existed before the pandemic, they became evermore apparent as stressors multiplied and demand could not be met.

**Family and individual strategies.**   Notwithstanding the important gaps identified at institutional level to access formal mental healthcare, participants displayed coping strategies at individual and family level, in the face of the health and socioeconomic crisis provoked by the pandemic.

The first and main strategy was managing their fear of the virus through information, following prevention recommendations and "learning how to live with the virus":

"No, I feel more settled now, I am not forgetting that I have to be careful, but I do not feel that same panic as before, that I might get ill tomorrow, or if one of my children gets ill and dies, I feel more settled, I have learned how to live with the virus"

Colombian woman, Santiago (3)

The other strategies described by the participants focused mainly on distracting themselves from the pandemic and lockdowns and keeping themselves busy at home, playing music, avoiding the news and interacting with their family and loved ones in Chile or in their home country:

"I have not listened to the news in almost 7 months, nothing, no news, only my husband watches the news, I don't, I prefer spending my time watching series to distract myself, but yes I felt scared, very scared"

Haitian woman, Santiago (1)

Finally, being able to go back to work eased anxieties related to the future and socioeconomic conditions:

"Now I feel better because I started working again"

Ecuadorean woman, Santiago (1)

Individual strategies relied on the resources participants had available to cope with the distress caused by the pandemic, but none reported negative coping strategies such as overdrinking or substance abuse.

## Discussion

The WHO, in its 2017 Draft framework of priorities and guiding principles to promote the health of refugees and migrants, establishes that the mental health of international migrants in the Americas is of special interest as regional migration flows increase, especially migrants in an irregular situation and migrants fleeing violence, with consequences to their mental health [48]. According to the International Organization for Migration, in 2020, South America was

home to nearly 11 million international migrants and mixed migration fluxes from the region and from the Caribbean have been on the rise [49,50]. International migrants are not inherently vulnerable to mental health issues; however, they face adverse events, or social vulnerability, which may alter their mental health outcomes [51].

Participants reported different degrees of adversity during the stages of their migratory cycle and some linked migration and mental health with regards to reasons for migrating or reporting symptoms of migratory grief after settling in Chile. Among potential stressors are having left their country of origin to flee violence, travelling to Chile by land or by foot and suffering discrimination and difficulties in finding employment. Migratory status was also an important underlying stressor with consequences on living conditions, well-being and mental health of the participants. This is consistent with the WHO report on mental health and migration [52] and the existing literature on migration as a social determinant of mental health [53–56]. More specifically, a quantitative study conducted in Peru with Venezuelan migrants found that walking at any point in their journey was associated with increased odds of anxiety, together with loss of status as a result of migration [57]. Additionally, some participants linked their mental health with their decision to migrate, something seldom described in the existing literature, which usually focuses on mental health during settlement or negative mental health outcomes due to the traumatic events that are themselves the reason to migrate. Mental health as a "push factor" has been described in in specific populations such as gay men suffering from place-based minority stress [58].

During the COVID-19 pandemic, international migrants in Chile reported both positive and negative mental health outcomes. Emphasizing positive outcomes among findings is key to avoid considering international migrants as a homogenously vulnerable group and to understand the social vulnerability experienced by migrants as fluctuating rather than inherent, modifiable with adequate intersectoral policies [59]. However, it is important to take into account that the participants reporting positive outcomes were all highly educated, with financial and migratory stability and, in most cases, had access to positive coping mechanisms, however, not all participants with these characteristics reported positive outcomes, especially when they only reported only one of them, and financial and migratory precariousness, usually interrelated, were important stressors. Individual determinants as well as social determinants linked to existing dynamics of exclusion based on class and race in Chile, as well as other determinants such as migratory status, socioeconomic status and educational level must, in that sense, be considered when examining the mental health of international migrants. The existing literature has identified additional protective factors for the mental health of international migrants, among which age, self-esteem, maintenance of cultural identity, social support, belonging and safety and access to innovative social care services [60]. Further research in the context of Latin American migrants in Chile needs to be conducted in order to identify protective factors and determine whether the ones identified in other contexts are also reproduced.

Notwithstanding positive outcomes, most participants reported negative mental health outcomes, such as anxious and depressive symptomatology. Stressors included dimensions directly related to the virus that also triggered stressors linked to being a migrant. Furthermore, as a result of the pandemic employment, living and socioeconomic conditions became interrelated daily stressors, as loss of work increased worries around personal and family finances or having to go to work during lockdowns increased feelings of vulnerability to the virus. These issues are not exclusive to international migrants, as the pandemic has brought about a social and economic crisis with catastrophic consequences especially for population experiencing social vulnerability [61], something that has also been identified in Chile [62]. However, international migrants face circumstances that exacerbate these stressors and bring about additional ones: precarious migratory status, discrimination as certain nationalities were

wrongfully singled out as spreading the virus, and dynamics of "distance worrying". The existing literature identifies similar stressors at global level [16–18,20,22–24] and it can be argued that international migrants can face compounded stress as a result of the different stressors encountered throughout their migratory cycle and of the COVID-19 pandemic, as has been described in the case of Venezuelan migrants in Colombia [63]. It is also important to note that the stressors identified stem from existing inequalities and dynamics of exclusion and discrimination, where racism, gender discrimination, and employment, living conditions and migratory precariousness are intertwined and exacerbated during a crisis such as the COVID-19 pandemic. It is also important to take into account that this reality is not always reflected, and a study conducted with Venezuelan women living in Peru during the pandemic found that the information broadcasted in the media was a source of preoccupation, fear and anger for them partly because the information was not relevant to their particular situation and did not address the issues they were facing as a result of COVID-19 [64].

With regards to the gender dimension of stressors during the pandemic, the central finding focuses on female live-in domestic workers, whose lives started revolving entirely around caring for others, leaving aside their personal lives. However, albeit a key aspect, the interviews did not capture evidence of mental health consequences of an additional burden of domestic work within their own home among the women interviewed, as found in other countries [65]. Further studies should explicitly include a gender approach when addressing the mental health of international migrants during the pandemic.

Institutional responses to mitigate mental health issues during the pandemic were limited, and gaps in access, availability and acceptability gaps were highlighted. This is consistent with the existing literature on mental healthcare in the context of international migration in Chile [33,66–68]. Conversely, civil society organisations developed short term solutions to promote urgent mental healthcare and a range of individual coping strategies were reported by the participants inside and outside health systems. More broadly, the lack of institutional support with regards to financial and material needs for vulnerable populations during the pandemic exacerbated the risk for negative mental health outcomes.

The pandemic can represent an important opportunity to strengthen mental health systems for the general population as well as for population groups experiencing social vulnerability, both in the short and long term [69], if the issues identified and the lessons learned are translated into action. First, it is important to consider the diversity of migrant groups at global, regional, and national level to avoid seeing them as inherently and invariably vulnerable to facing mental health issues. Contextualizing the mental health of international migrants, not in terms of increased mental health prevalence and care needs, but rather in terms of differentiated needs with regards to non-migrants but also within the migrant population as a heterogenous group, is key.

Promoting the mental health of international migrants means recognising migration as a social determinant of mental health and adopting a cross-cultural as well as a Human Rights approach. International migrants have the right to enjoy the highest possible standard of mental health and the right to access linguistically and culturally relevant mental healthcare according to their needs: not every international migrant will face mental health issues, however, health systems must ensure that those who do have the resources to cope and access care and support. Good practice identified in the existing literature on promoting the mental health of migrants includes information dissemination, active outreach, extensive case management to facilitate access to health insurance and other public services, telemedicine and online communication [70]. Additionally, we recommend reinforcing mental healthcare at primary care level, training support community teams to deliver psychological first aid and promoting support networks and peer support systems in communities where vulnerable international migrants live.

This study presents strengths and limitations. With regards to strengths, it is one of the few qualitative analyses on international migrants' mental health during the pandemic, bringing the perspective of migrants from Latin America and the Caribbean to Chile, in the context of South-South migration patterns. Additionally including a detailed sub-analysis of the migratory trajectory of the participants and its relation to mental health, our study connects migration as a social determinant of mental health and the consequences of the COVID-19 on the mental health of international migrants.

With regards to limitations, although care was taken in building a diverse sample, some population groups were left out due to feasibility issues: non-Spanish speaking international migrants could not be interviewed, neither were international migrants in detention nor held in quarantine facilities. The perspective of older migrants was not fully accounted for, and children were not included in the sample. Finally, an important limitation regarding the secondary analysis focused on mental health that is presented in this article is that mental health emerged as a topic of interest in the primary study rather than being its focus and in that sense, not all possible dimensions of migration and mental health during the pandemic may have emerged in this secondary analysis. Saturation was reached for the primary study, however, further in-depth studies on the mental health of international migrants in Chile and the Latin America region during the pandemic are needed.

## Conclusions

International migrants may experience different mental health issues to varying degrees depending on their pre-migratory conditions, pre-existing conditions, transit conditions, settlement conditions and during the pandemic, depending on their resources to cope with the consequences of the pandemic. In addressing the mental health of international migrants, it is important not to erase these nuances or remove their agency and overlook individual circumstances, resources, and coping strategies. However, further institutional support is urgently required to address mental health and this support must be intersectoral in order to address the multiple stressors international migrants face in the context of the pandemic and beyond.

Our study is relevant for policymakers in Chile and Latin America, as it informs the dimensions of mental health of international migrants, including migratory and mental health trajectories, symptomatology, stressors and institutional and individual strategies to address the consequences of the pandemic. Additionally, it calls for mental health of populations experiencing social vulnerability to be put at the centre of the pandemic response.

## Supporting information

**S1 Checklist.**
(PDF)

**S1 File. Interview guide.**
(PDF)

## Acknowledgments

The authors thank everyone who contributed to the project, including all participants.

## Author Contributions

**Conceptualization:** Báltica Cabieses, Alexandra Obach, Kate E. Pickett, Niina Markkula.

**Formal analysis:** Alice Blukacz, Paula Madrid.

**Funding acquisition:** Báltica Cabieses, Alexandra Obach.

**Methodology:** Báltica Cabieses, Alexandra Obach, Alejandra Carreño.

**Project administration:** Alice Blukacz.

**Supervision:** Báltica Cabieses, Alexandra Obach, Kate E. Pickett, Niina Markkula.

**Validation:** Paula Madrid, Alejandra Carreño.

**Visualization:** Alice Blukacz.

**Writing – original draft:** Alice Blukacz, Báltica Cabieses, Paula Madrid.

**Writing – review & editing:** Alexandra Obach, Alejandra Carreño, Kate E. Pickett, Niina Markkula.

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
