## [Decision Letter · Decision Letter 0]

16 Aug 2022

PONE-D-22-18476“If I get sick here, I will never see my children again”: the mental health of international migrants during the COVID-19 pandemic in ChilePLOS ONE

Dear Dr. Cabieses

Thank you for submitting your manuscript to PLOS ONE. After careful consideration, we feel that it has merit but does not fully meet PLOS ONE’s publication criteria as it currently stands. Therefore, we invite you to submit a revised version of the manuscript that addresses the points raised during the review process.

Although the opinions of reviewers are different, both agree that the work requires changes. On this occasion I agree with the second reviewer and concur that additional work is required to be able to publish this work. Please review the comments in detail and please put attention regarding the relevance of the research question thah guided the study and please do include a more detailed limitations section.  Another issues to consider are:

- How were verbatim quotes selected? please provide information and justify the quotes inserted.

- I suggest further developing the analysis that is made of the testimonies included and if possible incorporate some that better represent the topics that are discussed. Many are very short and do not present sufficient evidence. Additionally, I suggest not concluding the paragraphs with a testimony. Please develop the analysis after presenting a testimony.

- Eliminate the two initial tables that take up too much space and also have important wording and style details. Please perform an analysis even if it is descriptive of the sociodemographic information and include it in the development of the text. The tables take away a lot of seriousness and quality to the work. 

We look forward to receiving your revised manuscript.

Kind regards,

Cesar Infante Xibille, Ph.D

Academic Editor

PLOS ONE

Journal Requirements:

Reviewers' comments:

Reviewer's Responses to Questions

**Comments to the Author**

1. Is the manuscript technically sound, and do the data support the conclusions?

Reviewer #1: Yes

Reviewer #2: Partly

2. Has the statistical analysis been performed appropriately and rigorously? 

Reviewer #1: N/A

Reviewer #2: N/A

3. Have the authors made all data underlying the findings in their manuscript fully available?

Reviewer #1: No

Reviewer #2: No

4. Is the manuscript presented in an intelligible fashion and written in standard English?

Reviewer #1: Yes

Reviewer #2: Yes

5. Review Comments to the Author

Reviewer #1: This is a nice, well-written, descriptive article focusing on the mental health issues of migrants in Chile during the COVID-19 pandemic. I have only a few suggestions for the authors:

1) From the description of the migrants’ characteristics, they seem diverse in their sociodemographic characteristics. I suggest the authors add this to the analysis: Did socioeconomic position explained some of the outcomes observed? Could experiences of discrimination be different between migrants in higher or lower socioeconomic positions? What about intersectionality (differences by race, national origin, gender)?

2) Related to my previous comment: the finding that the pandemic resulted in positive mental health outcomes for some is very interesting, but maybe only the more privileged migrants reported positive outcomes. The quote supporting this theme is from an Argentinian man who was university-educated, had a valid residence permit and seemingly a good job. It may be that his ability to enjoy some aspects of the pandemic are related to his socioeconomic position. Did other participants report positive outcomes? Could the authors comment on this, perhaps by developing the phrase “Emphasizing positive outcomes among findings is key to avoid considering international migrants as a homogenously vulnerable group”?

3) That some interviewees reported mental health problems as a reason for migration is a very interesting finding. I suggest the authors discuss it a little bit more, as it is seldom addressed in studies of migration and mental health.

4) I suggest the authors discuss the similarities and differences between the negative mental health outcomes related by migrants, and what the non-migrant population was experiencing at the time. Fear of job loss and other social determinants probably were highly prevalent among other persons of lower socioeconomic position as well.

5) A minor issue: When exactly were the interviews conducted? This information seems missing from the article, and since the pandemic has been going one for a long time, it would be important to know if the interviews were conducted during lockdown periods, before or after vaccination, etc.

Reviewer #2: Thank you for the opportunity for reviewing this text. Is a very well written and organize article.

The following are recommendations that seek to improve the text.

1.There is no goal/objective or research question

2. Adding a discussion about how the secondary analysis methods were used is recommended. This is important because the interview guide does seem to be focused on mental health in any of its sections. Some texts that may be helpful:

Heaton, J. (2004). Secondary Analysis of Qualitative Data. Thousand Oaks, CA: Sage doi: 10.4135/9781446212165.n30.

Ruggiano, N., y Perry, T. E. (2019). Conducting secondary analysis of qualitative data: Should we, can we, and how? Qual Soc Work 18, 81–97. doi: 10.1177/1473325017700701.

Some questions to address: Did the authors did a recodification? How do they analyze mental health in the secondary study? How does categories or themes in where developed?

3. Authors mentioned that “Figure 1 describes the emerging categories, main categories, generic categories and subcategories from the analysis process as described in the methods section.” This is not clear in the figure.

4. The number of participants seems to be too high for a case study, for example case studies focus in generating in depth and extensive knowledge, which may not be the case as the collection methods implied 45 min long interviews. Explaining how the authors decided on “saturation” and including the research question may help to understand the study design.

Creswell, W. J., & Creswell, D. J (2018). Research design: Qualitative, quantitative, and mixed methods approach. Sage publications.

5. Also, to mention the sample criteria for the case study will be helpful especially due to the diversity of the participants (nationality, migratory status, sex, ages).

6. Within this diversity of participants, it would be important to address differences inside the sample (sex, country, migratory status) in the analysis and discussion.

7. Many of the reported stressors are not due only to the COVID-19 pandemic and this may have to be discussed, ex. discrimination.

8. The article may benefit from discussing with other Latin American literature on the subject.

6. PLOS authors have the option to publish the peer review history of their article (what does this mean?). If published, this will include your full peer review and any attached files.

Reviewer #1: No

Reviewer #2: No

---

## [Author Response · Author response to Decision Letter 0]

12 Oct 2022

Dear Dr Infante Xibille,

Many thanks for your decision on our manuscript and for kindly sharing with us the reviewers’ comments, which were very constructive and greatly helped us to revise our submission. We have made a range of specific changes, which are detailed in the tables in the file titled "Response to reviewers", responding to each of your comment as well as those made by the reviewers.

The manuscript is re-submitted with tracked changes.

Regarding the anonymised data from the study, we have made it available on the institutional repository of the Universidad de Desarrollo and it is accessible through the following link: http://hdl.handle.net/11447/6583. 

We hope that this new version of the manuscript will be suitable for publication.

Kind regards,

Báltica Cabieses, PhD (Corresponding author)

---

## [Decision Letter · Decision Letter 1]

31 Oct 2022

“If I get sick here, I will never see my children again”: the mental health of international migrants during the COVID-19 pandemic in Chile

PONE-D-22-18476R1

Dear Dr. Cabieses,

We’re pleased to inform you that your manuscript has been judged scientifically suitable for publication and will be formally accepted for publication once it meets all outstanding technical requirements.

Kind regards,

Cesar Infante Xibille, Ph.D

Academic Editor

PLOS ONE

Reviewers' comments:

Reviewer's Responses to Questions

**Comments to the Author**

1. If the authors have adequately addressed your comments raised in a previous round of review and you feel that this manuscript is now acceptable for publication, you may indicate that here to bypass the “Comments to the Author” section, enter your conflict of interest statement in the “Confidential to Editor” section, and submit your "Accept" recommendation.

Reviewer #1: All comments have been addressed

2. Is the manuscript technically sound, and do the data support the conclusions?

Reviewer #1: Yes

3. Has the statistical analysis been performed appropriately and rigorously? 

Reviewer #1: N/A

4. Have the authors made all data underlying the findings in their manuscript fully available?

Reviewer #1: Yes

5. Is the manuscript presented in an intelligible fashion and written in standard English?

Reviewer #1: Yes

6. Review Comments to the Author

Reviewer #1: I have no further comments for the authors. All my previous comments were addressed. I think this is a relevant article.

7. PLOS authors have the option to publish the peer review history of their article (what does this mean?). If published, this will include your full peer review and any attached files.

Reviewer #1: No

---

## [Editor Report · Acceptance letter]

16 Nov 2022

PONE-D-22-18476R1 

“If I get sick here, I will never see my children again”: the mental health of international migrants during the COVID-19 pandemic in Chile 

Dear Dr. Cabieses:

I'm pleased to inform you that your manuscript has been deemed suitable for publication in PLOS ONE. Congratulations! Your manuscript is now with our production department. 

Kind regards, 

on behalf of

Dr. Cesar Infante Xibille 

Academic Editor

PLOS ONE